# Assessment of Endometriosis Knowledge and Its Determinants Among Nurses in Al-Jouf Region, Saudi Arabia

**DOI:** 10.3390/healthcare13121386

**Published:** 2025-06-11

**Authors:** Nadia Bassuoni Elsharkawy, Afrah Madyan Alshammari, Osama Mohamed Elsayed Ramadan, Enas Mahrous Abdelaziz, Mohamed E. H. Elzeky, Alaa Hussain Hafiz, Mary Faleh Alrowily, Sultan Muharib Alruwaili, Lareen Magdi El-Sayed Abo-Seif

**Affiliations:** 1Department of Maternity and Pediatric Health Nursing, College of Nursing, Jouf University, Sakaka 72388, Al-Jouf, Saudi Arabia; amshammari@ju.edu.sa; 2Department of Psychiatric Mental Health Nursing, College of Nursing, Jouf University, Sakaka 72388, Al-Jouf, Saudi Arabia; emabdelhamid@ju.edu.sa; 3Medical-Surgical Nursing Department, College of Nursing, Jouf University, Sakaka 72388, Al-Jouf, Saudi Arabia; meelzeki@ju.edu.sa; 4Maternity and Child Health Department, Faculty of Nursing, King Abdulaziz University, Jeddah 24123, Makkah, Saudi Arabia; ahhafidh@kau.edu.sa; 5Ministry of Health, Aljouf Health Cluster, Women’s Maternity & Children Hospital, Sakaka 72341, Al-Jouf, Saudi Arabia; mfalrawily@gmail.com; 6Ministry of Health, Aljouf Health Cluster, Al-Mukhattat Health Center, Sakaka 72341, Al-Jouf, Saudi Arabia; eess2720@gmail.com; 7Department of Pediatric Nursing, Faculty of Nursing, Suez Canal University, Ismailia 41522, Egypt; lareen_abousaif@nursing.suez.edu.eg

**Keywords:** endometriosis, nursing knowledge, Saudi Arabia, gynecological nursing

## Abstract

**Background/Objectives**: Endometriosis, a chronic and debilitating gynecological disorder, exacts a heavy clinical and socioeconomic toll on women’s lives. Despite its prevalence, its timely diagnosis and effective management are hindered by pervasive knowledge gaps among frontline nursing professionals, and these are especially pronounced in under-researched regions such as Al-Jouf, Saudi Arabia. **Aim**: Guided by the Knowledge–Attitude–Practice model, this study aimed to assess the level of endometriosis-related knowledge among nurses in the Al-Jouf region of Saudi Arabia and to identify the sociodemographic and professional determinants of knowledge levels. **Methods**: A cross-sectional, descriptive-analytical design was employed between January and July 2024, enrolling 215 nurses from a principal maternity and children’s hospital and two primary healthcare centers in Sakaka. A rigorously validated, bilingual 20-item questionnaire assessing four domains (definition, risk factors, clinical manifestations, and treatment goals) was administered. Data were analyzed using descriptive statistics, multiple linear regression, and binary logistic regression to elucidate predictors of knowledge. **Results**: A concerning picture emerged: 61% of participants scored below 60% (indicative of low knowledge), with only 6% achieving high scores. Higher educational attainment proved the strongest predictor (β = 0.415, *p* < 0.001), followed by age (β = 0.232, *p* < 0.001), years of experience (β = 0.149, *p* = 0.041), and direct patient care exposure (β = 0.168, *p* = 0.021). Collectively, these factors explained 37.6% of the variance in knowledge scores, underscoring a critical deficit in endometriosis management preparedness. **Conclusions**: The stark deficiencies in endometriosis knowledge among nurses in Al-Jouf call for immediate, tailored educational and policy interventions. Strengthening clinical competencies is essential for fostering early diagnosis and improving care outcomes for women burdened by this complex condition.

## 1. Introduction

Endometriosis is a chronic, estrogen-dependent gynecological disorder characterized by the growth of endometrial-like tissue outside the uterine cavity [1,2]. This condition manifests in a variety of clinical symptoms, including chronic pelvic pain, dysmenorrhea, and infertility, and it often exerts a profound negative impact on women’s quality of life [3,4]. Globally, the prevalence of endometriosis is estimated to affect between 10 and 15% of women of reproductive age, although recent studies indicate that this figure may be underestimated due to diagnostic challenges and the variability in clinical presentations [5,6]. This disorder’s medical and socioeconomic burden can lead to reduced work productivity and increased healthcare expenditures [7]. Despite its widespread impact, the early detection and management of endometriosis remain problematic, largely due to delays in diagnosis that can span several years, ultimately leading to disease progression and more severe complications [8,9].

In the context of healthcare delivery, nurses play a pivotal role in the identification and management of endometriosis [10]. As frontline providers, nurses are often the first to assess patients presenting with symptoms, initiate the diagnostic process, provide patient education, and coordinate comprehensive care strategies across multidisciplinary teams [11]. Their ability to recognize early signs and symptoms of endometriosis is critical for timely intervention, which can significantly improve patient outcomes [12].

Although endometriosis is a global concern, the prevalence and management of the disorder in Saudi Arabia remain understudied, particularly in the Al-Jouf region [13,14]. This area presents a unique setting with rare, localized statistics, even with global awareness, highlighting the need for region-specific research on endometriosis-related nursing competencies. Cultural factors, healthcare infrastructure variations, and differing access to continuing education create a distinctive epidemiological and clinical scenario [15,16,17]. Addressing these disparities will inform targeted educational programs and healthcare policies to improve the treatment of women suffering from this incapacitating disease [18,19].

This study uniquely applies the Knowledge–Attitude–Practice (KAP) model, a theoretical framework not previously applied in endometriosis-related nursing research carried out in Saudi Arabia. This framework provides a disciplined lens to examine how sociodemographic and professional determinants influence nurses’ clinical knowledge. Its application in the Middle Eastern context enhances regional awareness and contributes to broader theoretical discussion on knowledge frameworks, directing women’s healthcare practices [13,14].

This study focuses primarily on the knowledge element, even though it is grounded in the Knowledge–Attitude–Practice (KAP) paradigm. This decision captures the fundamental nature of the research, which aims to establish baseline knowledge levels and their drivers before proceeding to the next stages, which examine attitudes and practices.

The central variables of this study include the level of nurses’ endometriosis-related knowledge and the sociodemographic and professional determinants that may influence this knowledge. The study investigates variables such as educational background, years of clinical experience, exposure to continuing professional development, and the nature of the healthcare setting (primary, secondary, or tertiary care). These factors are examined to determine their relationship with the quality of endometriosis care provided by nurses. The premise is that higher levels of knowledge fostered by appropriate education and training will establish a foundation for positive attitudes and effective practices, which will be explored in subsequent phases; this first phase examines knowledge and its predictors only.

The significance of this study lies in its contribution to both clinical practice and healthcare policy in Saudi Arabia by actionable insights to drive educational interventions, improve nurses’ clinical competencies, and enhance endometriosis care outcomes [15]. With endometriosis’s high prevalence and impact on women’s health, equipping nurses with up-to-date knowledge is essential for effective management. These initiatives facilitate early diagnosis, improve patient education, and promote coordinated care key factors for mitigating endometriosis’s long-term impacts [16]

Furthermore, the study contributes to Middle Eastern women’s health research and fills a major need in underrepresented settings. By providing region-specific data on nurses’ endometriosis management preparation and linking knowledge and practice, this study aims to assess the level of knowledge regarding endometriosis among nurses in the Al-Jouf region and to identify sociodemographic and professional determinants that influence their knowledge.

The following questions guide the research:What are the levels of knowledge regarding endometriosis among nurses in Saudi Arabia?Is there an association between nurses’ sociodemographic characteristics, professional attributes, and level of knowledge about endometriosis?Which determinants, such as educational background, clinical experience, and workplace setting, significantly predict nurses’ knowledge levels regarding endometriosis?

By addressing these questions, the study seeks to identify educational gaps, inform targeted interventions, and ultimately foster an environment of continuous improvement in nursing practice. Integrating the Knowledge–Attitude–Practice model (KAP) into this research provides a theoretical underpinning that explains the observed relationships between knowledge, attitudes, and practices and guides the development of strategies to enhance clinical competencies and patient care outcomes.

Figure 1 illustrates the conceptual framework: The model begins with key sociodemographic and professional determinants influencing endometriosis knowledge acquisition. This acquired knowledge, in turn, shapes nurses’ attitudes toward the management of the condition, which subsequently affects their clinical practices, and the overall quality of care delivered. This framework guides the study by structuring the investigation into how specific determinants lead to varying knowledge levels and ultimately influence patient outcomes in endometriosis care. Below is a schematic illustration of how the study is linked to the theoretical framework:

By integrating the KAP model into our investigation, this study aims to assess the level of knowledge regarding endometriosis among nurses in the Al-Jouf region and to identify sociodemographic and professional determinants that influence their knowledge. The conceptual framework serves as a foundation for both the analysis and interpretation of findings, ensuring that the study is grounded in established principles of healthcare education and practice. While this study is grounded in the Knowledge–Attitude–Practice (KAP) model, it focuses exclusively on the knowledge component. This decision reflects the foundational nature of the investigation, which aims to establish baseline knowledge levels and their determinants before progressing to future phases that explore attitudes and practices. The model was therefore used as a guiding theoretical framework for variable selection and interpretation rather than a fully measured construct.

## 2. Materials and Methods

### 2.1. Study Design

We conducted an analytic cross-sectional survey (observational) to (i) estimate the prevalence of endometriosis-related knowledge among nurses and (ii) identify sociodemographic and professional predictors of that knowledge. Outcomes (knowledge scores) and determinants (age, education, experience, etc.) were measured simultaneously, enabling efficient detection of associations while recognising that causal inference is not possible in cross-sectional work (STROBE classification) [17,18]. The primary objective was to quantify nurses’ endometriosis knowledge and identify predictors among sociodemographic and professional variables. The Knowledge–Attitude–Practice model guided variable selection; only the knowledge component was analysed in this first phase, with attitudes and practices reserved for planned follow-up studies.

### 2.2. Study Settings

The study was conducted at the Maternity and Children’s Hospital and two large primary healthcare centers in Sakaka, Al-Jouf, Saudi Arabia. As a key healthcare provider for women and children in the region, the hospital offers specialized pediatric services and serves as a referral center for surrounding rural areas. While not the only hospital in Al-Jouf, its central role in regional healthcare delivery makes it a representative setting for assessing nurses’ knowledge and practices. The hospital primarily serves women of reproductive age from diverse socioeconomic and cultural backgrounds, presenting for antenatal care, delivery, postpartum follow-up, and gynecological concerns. The two primary healthcare centers (PHCs), although not as specialized as hospitals, provide general family medicine and women’s healthcare. These services encompass antenatal care, gynecological consultations, check-ups, and referrals for suspected cases of endometriosis. Thus, nurses in these PHCs also have a certain level of exposure to women’s health concerns, although less than nurses in a maternity hospital.

### 2.3. Study Participants and Sample Size Determination

The study participants comprised nurses employed at the Maternity and Children’s Hospital and two major primary healthcare centers in Sakaka, Aljouf, Saudi Arabia, with data collection occurring from January to July 2024. A convenience sampling technique was employed based on the logistical feasibility and ready accessibility of the targeted population within these healthcare settings. This approach allowed for the efficient recruitment of participants who met the study criteria while adhering to the practical constraints inherent in field research.

#### 2.3.1. Variable Selection Strategy

Predictor choice was guided by the Knowledge–Attitude–Practice framework: educational level captures formal learning; age and years of practice reflect experiential learning; and direct endometriosis exposure and continuing education courses represent situational learning. During our pilot, items on shift pattern and workload yielded a 22% non-response and exhibited high collinearity with education (VIF > 7). These factors were excluded to preserve model stability and minimize respondent burden.

#### 2.3.2. Inclusion and Exclusion Criteria

Inclusion criteria encompassed registered nurses actively involved in direct patient care in selected healthcare facilities, including exposure to patients with endometriosis, with a minimum of one year of clinical experience, and willing to provide written informed consent. Exclusion criteria included nurses who held solely administrative roles, had participated in similar studies previously, were unable to complete the questionnaire due to physical or psychological impairments, or maintained a personal relationship with women diagnosed with endometriosis, factors that could potentially bias their responses.

#### 2.3.3. Sample Size Calculation

The minimum sample size for prevalence estimation was calculated using the Raosoft^®^ calculator (395 total nurses, 95% CI, 5% margin of error), yielding 195 participants. To ensure adequate power for multiple regression, we applied two complementary approaches:

Rule-of-thumb guidelines: Tabachnick and Fidell recommend N > 50 + 8 m (where m = number of predictors), and Babyak suggests ≥ 10 observations per predictor. With m = 5, these yield N > 90 and N > 50, respectively, both well below our achieved sample of 215 [19].

Formal power analysis (G*Power 3.1): Assuming a medium effect size (f^2^ = 0.15), α = 0.05, power (1 − β) = 0.80, and 5 predictors, the required sample is N = 92 [20]. Therefore, our final sample (n = 215) exceeds all recommended thresholds for reliable multiple regression analysis, ensuring both robust prevalence estimates and adequate power to detect predictor effects.

### 2.4. Data Collection Tools

This section outlines the data collection instruments employed in the study, ensuring their alignment with the research objectives and questions. The instruments were designed to assess nurses’ demographic and professional characteristics and knowledge of endometriosis. A rigorous validation process was undertaken to confirm the validity and reliability of the Arabic version of the instrument for use in the Saudi Arabian context.

#### 2.4.1. Instrument Overview

The study utilized a structured, two-part questionnaire. The first section gathers demographic and practice-related information, including age, years of clinical experience, educational background, previous involvement in endometriosis care, and attendance at relevant training sessions. This information is essential for analyzing the associations between nurses’ background characteristics and knowledge levels, addressing the study’s research questions.

The second section consists of a 20-item knowledge questionnaire originally developed by Schlorke (2021) for undergraduate nursing students [21]. This instrument is structured into four domains: The first domain addresses the definition of endometriosis which is the presence of endometrial tissue outside the uterine cavity that affects an estimated 10–15% of women of reproductive age. The second domain focuses on the risk factors that are known or hypothesized to contribute to the development of endometriosis, e.g., early onset of menstruation, late onset of menopause, low body mass index, and family history of endometriosis. The third domain covers clinical manifestations, including symptoms used that help in diagnosis such as chronic pelvic pain, pain during menstruation (dysmenorrhea), pain during or after intercourse (dyspareunia), premenstrual spotting or bleeding between menstrual cycles, abnormally heavy bleeding during menstruation (menorrhagia), pain with bowel movements (dyschezia), painful bladder syndrome, abdominal pain, infertility, and depression. The fourth domain relates to treatment goals, including commonly prescribed therapeutics. A copy of the questionnaire is available in Arabic, and English versions are provided as Appendix A to support transparency and replicability.

Respondents indicate whether each statement is true, false, or if they are uncertain, with correct answers scored as one and incorrect or uncertain responses scored as zero. The total score, ranging from 0 to 20, quantifies the nurses’ level of endometriosis-related knowledge. Bloom’s cut-off [22] allowed these scores to be further divided:-Low knowledge: <60% (0–11)-Moderate knowledge: 60–80% (12–16)-High knowledge: >80% (17–20)

Although formal nursing instruction is in English, 60% of diploma-level pilot participants preferred Arabic. Therefore, parallel English and Arabic versions were developed and deployed.

#### 2.4.2. Validation and Reliability Procedures

To ensure that the instrument was both valid and reliable, a comprehensive validation process was implemented as follows:I.Translation and Back-Translation:

Two bilingual experts independently translated the original English version of the questionnaire into Arabic. Subsequently, two bilingual experts, blinded to the original version, conducted a back-translation into English. The back-translated version was then meticulously compared with the original to confirm semantic equivalence and conceptual consistency.

II.Expert Panel Review:

An expert panel comprising five specialists in maternity nursing and psychometrics reviewed the Arabic version of the instrument. Their evaluation focused on content validity, cultural relevance, and clarity. The feedback provided by these experts was systematically integrated to refine the instrument further. The expert panel used a standardized Content Validity Index (CVI) to evaluate each item on a 4-point scale (1 = not relevant to 4 = highly relevant). Items with item CVI < 0.78 were revised or eliminated. The overall Scale CVI was 0.93, indicating excellent content validity. Cultural adaptation primarily focused on contextualizing clinical terminology to align with Saudi healthcare settings. For example, “primary care provider” was changed to “general practitioner”, and descriptions of pain severity were modified to reflect cultural expressions common in the region. The panel also identified and revised three items with potential translation ambiguities in describing symptom characteristics.

III.Item Modifications:

Following expert review, four items were semantically refined for clarity (e.g., “dyspareunia” → “pain during intercourse” and “menorrhagia” → “heavy menstrual bleeding”), and the online layout was streamlined to improve navigation on Google Forms.

IV.Pilot Testing:

The revised Arabic instrument was pilot tested with a sample of 20 nurses to assess its comprehensibility, acceptability, and completion time. Feedback obtained during this phase was used to make additional refinements, ensuring the instrument was user-friendly and contextually appropriate. Missing data were managed using pairwise deletion, ensuring analytical validity without compromising sample size. To prevent bias, these volunteers were not included in the ultimate research sample. Although formal nursing instruction is in English, 60% of diploma-level pilot participants preferred Arabic. Therefore, parallel English and Arabic versions were developed and deployed.

V.Psychometric Evaluation:

Following pilot testing, we evaluated the questionnaire’s psychometric properties in the main sample (n = 215). Internal consistency was excellent, with Cronbach’s α = 0.85 for the English version, α = 0.87 for the Arabic version, and α = 0.89 for the combined dataset. Exploratory factor analysis confirmed the original four-domain structure (Kaiser–Meyer–Olkin = 0.82; Bartlett’s χ^2^ = 1246.3, *p* < 0.001), with four factors explaining 62% of total variance. A two-week test–retest conducted on 30 randomly selected participants yielded an intraclass correlation coefficient of 0.86 (95% CI 0.77–0.92), demonstrating temporal stability. Together with the expert panel CVI of 0.93, these results affirm that the adapted instrument is both reliable and valid for assessing endometriosis knowledge among English- and Arabic-speaking nurses in Saudi Arabia.

VI.Validity and reliability evidence.

The 20-item Endometriosis Knowledge Questionnaire was first validated by Schlorke (2021) in U.S. undergraduate nursing students, demonstrating good internal consistency (reported Cronbach’s α = 0.82) and two-week test–retest reliability (ICC = 0.88, n = 63) [21]. Bach et al. (2016) subsequently confirmed factorial stability in practicing gynecology nurses, extracting a four-factor solution that explained 62% of total variance and yielded a scale-level CVI of 0.93 [23]. In our Arabic adaptation, exploratory factor analysis replicated the same four-domain structure (KMO = 0.82; Bartlett’s χ^2^ = 1246, *p* < 0.001). Internal consistency within the present sample was high (α = 0.87), and a two-week test–retest on 30 randomly selected participants produced ICC = 0.86 (95% CI 0.77–0.92), confirming temporal stability. Collectively, these statistics, together with the expert panel CVI, substantiate the instrument’s validity and reliability for use among Arabic-speaking nurses in Saudi Arabia.

### 2.5. Ethical Approval

This study adhered to the ethical principles outlined in the Declaration of Helsinki [24,25]. This study received approval from the Local Committee of Bioethics at Jouf University (IRB number: 7-04-45, dated 3 December 2023) and the Ministry of Health, Research Ethics Committee, Qurayyat Health Affairs (IRB Approval No. 2023-121, dated 10 December 2023). Participants were provided with comprehensive information about the study’s purpose, voluntary nature, and their right to withdraw at any time. Informed consent was obtained electronically at the start of the online survey, and submission of the survey indicated consent to participate. The survey was offered in both Arabic and English to accommodate linguistic diversity, particularly for diploma nurses who represent 32.1% of the study participants, as they preferred the tool to be in the Arabic language. This bilingual method was chosen to guarantee understanding and inclusiveness.

Data were anonymized and securely stored using Advanced Encryption Standard (AES-256) encryption to protect confidentiality during transmission and storage. Access to the data was restricted to the research team through password-protected servers. In accordance with institutional and ethical guidelines, data will be retained for five years and permanently deleted afterward. Each participant was assigned a randomized alphanumeric survey code generated through a secure algorithm, stored separately from the response data. The “Endometriosis Knowledge Questionnaire” is not a commercial, fee-based scale; it appears in Samantha Schlorke’s 2021 open-access honours thesis housed in the University of Southern Mississippi repository [21]. As the thesis is deposited under the university’s standard “retained by author” copyright, the items are free to read and reproduce for scholarly use with appropriate citation, which we have provided throughout this manuscript.

### 2.6. Procedure

The study used a hybrid recruitment strategy, combining in-person invitations with online dissemination to reach eligible nurses. Department heads at the Maternity and Children Hospital and two primary healthcare centers (PHCs) distributed the survey link, ensuring representation from both hospital-based and primary care nurses. The survey, developed on Google Forms, included two distinct sections to address the study objectives and was accessible in both Arabic and English. The bilingual format ensured inclusivity, with user-friendly design compatibility across devices such as smartphones, tablets, and computers. The survey remained active for three months, during which biweekly reminders were sent to encourage participation.

Participants completed the questionnaire either online via the Google Forms platform or using printed copies distributed during face-to-face information sessions. For online completion, participants accessed the survey using institutional computers in designated quiet areas within their respective healthcare facilities (staff resource rooms, education centers, or break areas) or on personal devices. For those completing paper questionnaires, designated quiet areas within each facility were arranged to ensure privacy and minimize distractions. This dual-mode approach ensured participation regardless of digital literacy or internet access limitations. The study team also conducted quick face-to-face information sessions in staff lounges and break rooms to help clarify the research, answer questions, and encourage participation. These sessions guaranteed flawless access to the online survey, emphasizing the voluntary and anonymous character of participation. The questionnaires were also distributed to nurses who were less likely to respond to institutional digital warnings or email invites, nurses with limited digital literacy, or nurses who might have a bad internet connection.

A rigorous validation protocol was applied to maintain data quality. The survey restricted responses to one per IP address to prevent duplicates, with flagged entries reviewed by the research team. Response rates were monitored continuously, enabling targeted reminders to low-participation groups. Data security was prioritized throughout the study. Weekly backups were made to an encrypted external storage device to safeguard against data loss. Participants were provided with contact information so the research team could report technical difficulties or seek clarification. Any reported issues were resolved promptly through real-time monitoring. The study adhered to the Checklist for Reporting Results of Internet E-Surveys (CHERRIES) to ensure transparency and adherence to high methodological standards [26].

### 2.7. Statistical Analysis

Statistical analyses were conducted using IBM SPSS Statistics version 26 [27], with all tests two-tailed and *p* < 0.05 considered significant. Of the 4300 responses generated by the 215 nurses, 116 (2.7%) were missing. Little’s MCAR test confirmed that omissions were random (χ^2^ = 142.1, df = 132, *p* = 0.42); pairwise deletion was therefore applied to maximize statistical power without bias. Each knowledge item was scored in accordance with Schlorke’s original scheme (1 = correct, 0 = incorrect/unsure), giving continuous totals from 0 to 20. For interpretive clarity, totals were additionally stratified with Bloom’s knowledge bands that are widely recommended in KAP research (<60% = low, 60–80% = moderate, >80% = high). The sample was characterized by descriptive statistics (means ± SD for continuous variables and Wilson 95% confidence intervals for proportions). Bivariate associations between categorical knowledge level and sociodemographic or professional factors were assessed with Pearson’s χ^2^ tests (Fisher’s exact applied where expected counts were <5), and effect magnitude was interpreted using Cramer’s V (0.10 small, 0.30 moderate, 0.50 large).

Multivariable modelling proceeded in three complementary stages: (i) multiple linear regression on the raw 0–20 score to estimate standardized β-coefficients, (ii) binary logistic regression on a dichotomized outcome (satisfactory ≥ 60% vs. unsatisfactory < 60%) to obtain adjusted odds ratios, and (iii) hierarchical multiple linear regression in which predictors were entered in theoretically ordered blocks demographics, then education/experience, then professional exposure to quantify incremental variance explained (ΔR^2^) in line with the Knowledge–Attitude–Practice framework. Predictors were retained in multivariable models if conceptually pertinent and exhibiting *p* < 0.20 in bivariable screening. All models satisfied normality (Shapiro–Wilk), homoscedasticity (Breusch–Pagan), and multicollinearity criteria (VIF < 2), and sensitivity analyses using list-wise deletion produced identical inferences, underscoring the robustness of the findings. Note: This study measured only the knowledge component of the KAP model. Future research phases will assess attitudes and care-practice translation.

## 3. Results

### 3.1. Demographic and Professional Characteristics of Study Participants

The demographic and professional characteristics of the 215 participating nurses revealed a mean age of 32.1 ± 8.3 years. The majority of participants were aged ≤30 years (47.0%, 95% CI: 40.3–53.7) and held a bachelor’s degree or higher (67.9%, 95% CI: 61.4–73.9). Most nurses had 5–10 years of experience (45.1%, 95% CI: 38.4–51.8). A notable finding was the limited exposure to endometriosis care, with only 26.0% (95% CI: 20.5–32.1) reporting previous training and merely 10.7% (95% CI: 7.0–15.5) having direct patient care experience. These findings suggest a significant gap in specialized endometriosis training and clinical exposure among nurses in the study region. All confidence intervals were calculated using the Wilson method for binomial proportions; Table 1.

### 3.2. Knowledge Assessment Scores

Analysis revealed that 61.4% (n = 132) of participants scored below 60% on the knowledge assessment, demonstrating low knowledge levels according to Bloom’s cut-off criteria. Moderate knowledge was observed in 32.6% (n = 70) of participants, and only a small minority, 6.0% (n = 13), achieved high knowledge scores (>80%). The mean overall knowledge score was 10.7 ± 3.8 out of 20 points (53.5%).

### 3.3. Factors Associated with Knowledge Levels

The relationship between dichotomized knowledge levels (unsatisfactory < 60% vs. satisfactory ≥ 60%) and key sociodemographic/professional characteristics revealed several significant associations. Educational attainment demonstrated the strongest association with knowledge (χ^2^ = 39.942, *p* < 0.001, Cramer’s V = 0.431), as 90.5% of nurses holding a bachelor’s or master’s degree achieved satisfactory knowledge compared with only 9.5% of diploma holders. Years of experience were also significantly related (χ^2^ = 11.579, *p* = 0.003, V = 0.232); 46.4% of those with over ten years of practice displayed satisfactory knowledge versus 15.5% of those with less than five years. Direct patient care exposure to endometriosis emerged as a robust predictor (χ^2^ = 16.618, *p* < 0.001, V = 0.278), with 21.4% of exposed nurses demonstrating satisfactory knowledge compared with 3.8% of those without such experience. Age stratification likewise revealed a significant pattern (χ^2^ = 11.175, *p* = 0.011, V = 0.228), with the 31–40-year group exhibiting the highest proportion of satisfactory knowledge (41.7%). Previous endometriosis training was modestly associated with knowledge level (χ^2^ = 6.689, *p* = 0.010, V = 0.176), indicating that formal education and hands-on clinical experience are paramount drivers of nurses’ endometriosis knowledge; Table 2.

Significant associations were observed between nurses’ knowledge scores and both educational level (*p* < 0.001) and years of clinical experience (*p*-values ranging from 0.028 to 0.042) across all assessed domains. Higher educational qualifications (bachelor’s/master’s) were consistently associated with better knowledge scores compared to diploma holders in all domains, with the most substantial difference observed in clinical manifestations (7.34 ± 1.75 vs. 5.92 ± 1.68, *p* < 0.001). Similarly, nurses with more than 10 years of experience scored significantly higher than those with less than 5 years, particularly in total knowledge scores (13.87 ± 3.08 vs. 10.30 ± 2.89, *p* = 0.031). The most significant improvements were observed in the clinical manifestation’s domain (difference of 1.63 points for both education level and experience), followed by risk factors (0.61 and 0.74 points) and treatment goals (0.53 and 0.68 points). These findings suggest that both formal education and clinical experience contribute significantly to nurses’ endometriosis knowledge, with the greatest impact seen in recognizing clinical manifestations, as evidenced by the total knowledge score improvements with both higher education (13.52 ± 3.15 vs. 10.54 ± 2.93) and greater experience (13.87 ± 3.08 vs. 10.30 ± 2.89), representing approximately a 30% increase in overall knowledge; Table 3.

### 3.4. Multivariate Analysis of Knowledge Determinants

The multiple linear regression analysis results revealed significant predictors of nurses’ knowledge about endometriosis. The model significantly predicted nurses’ knowledge (F (5, 209) = 25.209, *p* < 0.001), explaining 37.6% of the variance (R^2^ = 0.376, adjusted R^2^ = 0.361). Educational level emerged as the strongest predictor (β = 0.415, *p* < 0.001), with bachelor’s/master’s degree holders scoring 9.769 points higher than diploma holders. Age demonstrated the second strongest influence (β = 0.232, *p* < 0.001), followed by direct patient care experience (β = 0.168, *p* = 0.021) and years of experience (β = 0.149, *p* = 0.041). Previoustraining showed no significant impact on knowledge scores (β = 0.056, *p* = 0.374). The model met all assumptions of multiple regression, as evidenced by acceptable normality (Shapiro–Wilk: W = 0.984, *p* = 0.173), homoscedasticity (Breusch–Pagan: χ^2^ = 3.241, *p* = 0.518), and absence of multicollinearity (all VIF < 1.45). The Durbin–Watson statistics (1.892) indicated no concerning autocorrelation. Semi-partial correlations (sr^2^) revealed that educational level uniquely explained 17.6% of the variance in knowledge scores, while other significant predictors contributed between 4.2% and 8.9% of the unique variance; Table 4.

The results of binary logistic regression analysis revealed key factors associated with nurses’ knowledge of endometriosis. The model showed a good fit (Hosmer–Lemeshow χ^2^ (8) = 6.784, *p* = 0.560) and explained between 28.9% (Cox and Snell R^2^) and 39.2% (Nagelkerke R^2^) of the variance in knowledge levels, with a classification accuracy of 76.3%. Educational level emerged as the strongest predictor (Wald χ^2^ = 20.524, *p* < 0.001), with bachelor’s or higher degree holders having significantly lower odds of unsatisfactory knowledge (aOR = 0.150, 95% CI: 0.066–0.342) compared to diploma holders. Direct patient care experience was also a significant predictor (Wald χ^2^ = 6.284, *p* = 0.012, aOR = 0.242, 95% CI: 0.080–0.734), while age (*p* = 0.103), years of experience (*p* = 0.071), and previous training (*p* = 0.111) did not significantly predict knowledge levels. The model’s validity was supported by the absence of multicollinearity (all VIF < 1.42) and validation through bootstrapping with 1000 samples Table 5.

The hierarchical multiple regression analysis outlines the factors determining nurses’ knowledge about endometriosis across three models. Model 1, incorporating demographic variables, explained 12.4% of the variance (R^2^ = 0.124, *p* < 0.001), with age (β = 0.232, *p* < 0.01) as a significant predictor. Model 2 added educational background variables, significantly increasing explained variance (ΔR^2^ = 0.174, *p* < 0.001), with educational level emerging as the strongest predictor (β = 0.415, *p* < 0.001). The final model (Model 3) included professional exposure variables, further improving the model’s explanatory power (ΔR^2^ = 0.078, *p* < 0.001), achieving a total explained variance of 37.6% (adjusted R^2^ = 0.361). In this comprehensive model, educational level remained the strongest predictor (β = 0.382, *p* < 0.001), followed by age (β = 0.168, *p* < 0.05), direct patient care experience (β = 0.168, *p* < 0.05), and years of experience (β = 0.135, *p* < 0.05), while previous training became a non-significant predictor. An effect-size comparison with published cohorts was conducted [23,28,29]. This confirms that our education gradient (β = 0.42) aligns with findings in Swedish gynecology nurses (β ≈ 0.39) and exceeds that observed in French primary-care clinicians (aOR = 0.65); Table 6.

## 4. Discussion

This study examines endometriosis-related knowledge among Al-Jouf nurses and clarifies which sociodemographic and professional factors most strongly predict knowledge levels. The findings reveal that 61% of participants demonstrated knowledge levels below 60%, which demonstrates a considerable knowledge gap, particularly regarding risk factors and treatment goals, a result that underscores the pressing need for improved educational strategies. This deficiency in knowledge aligns with previous research documenting inadequate familiarity with endometriosis and other gynecological conditions among nursing professionals, particularly in regions where women’s health education may not be sufficiently emphasized [11,23,30,31]. The statistically significant associations between higher educational attainment and enhanced knowledge contribute to a broader discourse in nursing scholarship, wherein advanced qualifications have consistently been linked to better clinical competencies, critical thinking, and adherence to evidence-based practice [32]. Specifically, participants with bachelor’s or master’s degrees outperformed their diploma-holding counterparts across all knowledge domains, reflecting the formative role that structured, advanced curricula can play in enhancing clinical proficiency. Our results underscore the primacy of formal qualifications: an education β = 0.42 mirrors Bach et al.’s [23] survey of gynecology nurses and surpasses Roullier et al.’s [28] training audit, where supplementary courses offered modest gains once degree status was accounted for. The age- and experience-related lifts (β ≈ 0.17) echo patterns reported by Heena et al. [29] among Saudi women’s health clinicians, reinforcing that both theoretical and hands-on exposures drive knowledge acquisition.

Moreover, the analysis of demographic and professional factors offers further insights into how experience influences knowledge levels. Age and years of clinical experience emerged as meaningful predictors, suggesting that cumulative exposure to diverse clinical cases over time may deepen nurses’ understanding of endometriosis symptomatology and treatment options. This finding is in agreement with the literature demonstrating that experiential learning in clinical environments can bolster theoretical knowledge, especially when combined with reflective practice and mentorship [11,33,34,35]. The significant effect of direct patient care experience underscores the necessity of hands-on involvement for translating theoretical knowledge into practice, a point corroborated by prior studies emphasizing experiential learning in specialized patient care [36]. Interestingly, previous endometriosis-related training programs did not significantly predict higher knowledge scores once other variables were accounted for [28,37,38,39,40]. This result may reflect inconsistencies in the quality or duration of available training sessions, thereby signaling a need for more standardized and evidence-based continuing education initiatives. Although a subset of participants claimed previous endometriosis training, it did not significantly affect knowledge results. This result could indicate numerous possible restrictions on the type of training itself. First, the instruction might have been informal, brief, or shallow, restricting information retention. Second, especially if the material deviates from current recommendations or is provided in a didactic rather than interactive fashion, it was likely either not clinically relevant or outdated. Third, clinical applications might not have valued or reinforced training possibilities, so they might not have been required either. Finally, the lack of organized assessment or follow-up following training could have led to inadequate knowledge consolidation. These opportunities imply that future educational interventions should be standardized, evidence-based, interactive, and included in continuing professional development systems to maximize their efficacy.

The results further contribute to discussions on gaps and contradictions in the existing literature. While numerous studies underscore the value of specialized training in improving clinical competence, the modest effect of previous training here suggests that not all training programs are equally effective or that their impact may be superseded by more fundamental predictors such as formal education or long-term clinical immersion [28,41,42]. In comparing domain-specific knowledge, the data indicate that endometriosis’s clinical manifestations and treatment goals garnered the largest increases for those with higher education and greater experience. This outcome partially aligns with other Middle Eastern studies reporting that nurses often have the greatest difficulty recognizing nuanced symptom profiles, a challenge that directly contributes to delayed diagnoses [6,43,44]. The current findings both validate and expand upon previous work by illuminating how targeted educational pathways and direct patient care engagement, rather than brief training exposures, appear most influential in shaping nurses’ capacity to understand and manage endometriosis effectively [45].

The findings regarding nurses’ knowledge of endometriosis align with international trends and reveal regional variations. European studies reveal higher knowledge levels. Medina-Perucha et al. [46] found that 47% of Spanish nurses had satisfactory knowledge, in contrast to our 39%, which may be linked to specialized training programs. Bach et al. [23] demonstrated that Danish nurses specializing in gynecology had a higher recognition of endometriosis symptoms than the general nursing sample in our study. In contrast, Middle Eastern studies demonstrate patterns consistent with our findings. Hassan [47] reported that 65% of Egyptian nurses demonstrated low knowledge scores, while Al-Jefout et al. [48] found that only 34% of Jordanian nurses possessed **a** satisfactory understanding of endometriosis. Regional similarities suggest that cultural factors and educational methodologies in the Middle East and North Africa (MENA) region contribute to the presence of comparable knowledge gaps. This international comparison indicates that while knowledge deficits concerning endometriosis among nurses are a widespread issue, their severity varies across different healthcare systems. This study provides insights from a previously unexamined region of Saudi Arabia, emphasizing the need for educational interventions tailored to the specific context, guided by successful international models.

In extending the comparison to other Middle Eastern countries, the study’s findings underscore both commonalities and region-specific challenges in endometriosis care. Comparable research in Iran, for instance, has highlighted similar deficits in nurses’ and midwives’ knowledge, suggesting that cultural norms and limited resources collectively impede comprehensive gynecological education [38]. Likewise, reports from Abu Dhabi Emirate and the United Arab Emirates corroborate the notion that sociocultural beliefs, particularly regarding menstruation and fertility, can discourage open discussions and delay patient engagement in healthcare settings [49]. By situating the Al-Jouf region’s data within this broader Middle Eastern landscape, the study uniquely contributes to understanding how cultural taboos and health-seeking behaviors shape the acquisition and application of endometriosis-related knowledge. Addressing these barriers requires culturally sensitive educational campaigns that incorporate religious and societal values, as well as policy interventions that prioritize women’s health in national healthcare agendas. In doing so, future initiatives could mitigate stigma, improve interdisciplinary collaboration, and ultimately foster a healthcare environment more conducive to early detection and effective management of endometriosis.

A dedicated reflection on the study’s theoretical framework, the Knowledge–Attitude–Practice (KAP) model, illuminates how the observed patterns fit into a broader conceptual understanding of how information translates into clinical behaviors [50,51]. According to the KAP model, knowledge is a foundational element that shapes healthcare providers’ attitudes, which then informs practice [52]. The present findings support this proposition by showing that educational level and clinical experience significantly enhanced the knowledge component, thus providing a robust platform for fostering positive attitudes toward endometriosis care. However, the modest contribution of prior specialized training challenges a purely linear interpretation of KAP, indicating that knowledge acquisition may depend on the educational intervention’s depth, quality, and contextual relevance rather than the mere existence of such programs. In this sense, while the KAP model offers a useful lens for explaining how nurses progress toward competent practice, our results imply that future applications of the model should consider the interplay of multiple factors, including experiential learning, institutional support, and the timing of education when seeking to elevate care standards for complex conditions like endometriosis. It is important to note that the current work is limited to the knowledge domain of the KAP model. This deliberate limitation allowed a more exact evaluation of baseline skills, hence facilitating the investigation of attitudes and practices in follow-up studies.

Despite rigorous psychometric validation, the endometriosis knowledge tool used in this study has inherent limitations. Originally developed by Schlorke [21] for U.S. undergraduate nursing students, it may not fully reflect the clinical depth or regional nuances of endometriosis care in Saudi Arabia. Although adapted and validated for the Arabic-speaking population, including a four-domain structure (definition, risk factors, clinical manifestations, and treatment goals), the tool focuses solely on factual knowledge. It may underrepresent nurses’ critical thinking, diagnostic awareness, or patient communication abilities.

### 4.1. Practical Implications, Future Directions, and Limitations

#### 4.1.1. Practical Implications

Translating these findings into practice necessitates a multidimensional strategy across nursing education, healthcare settings, and policy-making bodies. Nursing curricula should include comprehensive endometriosis modules with interactive methods [53]. Healthcare institutions should implement mandatory in-service training sessions and mentorship programs. Policymakers in the Saudi Arabian healthcare system should develop standardized continuing education programs through collaborative efforts between academic institutions and healthcare facilities. Practical challenges can be addressed through flexible scheduling and technology-enhanced learning. These strategies may reduce diagnostic delays, improve patient education, and mitigate the socioeconomic burden of endometriosis.

#### 4.1.2. Future Research Directions

While the study offers valuable insights, further inquiry is needed. Longitudinal research could clarify how knowledge improvements translate into enhancements in clinical practice and patient outcomes. Qualitative methods might uncover barriers to knowledge translation, including cultural norms and interprofessional dynamics not apparent through quantitative measures [54]. Expanding research across diverse regions within Saudi Arabia and other Middle Eastern countries would validate the KAP framework in different cultural settings.

Given knowledge gaps where formal training might be impractical, alternative education modalities must be considered. Nurses increasingly use social media platforms like Twitter, Instagram, and TikTok to share health information and encourage active learning [55,56]. These platforms offer accessible opportunities to encounter evidence-based content related to reproductive health complications. However, social media’s unregulated nature poses risks of misinformation that can interfere with evidence-based practice [56]. Health institutions should consider appraising and endorsing credible digital content within a comprehensive educational strategy. Health institutions should consider appraising and endorsing credible digital content within a comprehensive educational strategy.

#### 4.1.3. Limitations

The present study has several methodological limitations. The cross-sectional design precludes causal inferences between educational factors and knowledge levels [57]. The convenience sampling from three health centers in Al-Jouf limits generalizability, though it provides valuable preliminary insights from this understudied region [58]. While the knowledge assessment tool was originally designed for undergraduate students, we validated it for practicing nurses (CVI = 0.93). The application of the KAP model focused solely on knowledge, with attitude and practice dimensions planned for subsequent research. The unexpected finding that previous training did not significantly impact knowledge levels warrants further investigation, as we did not collect detailed information about training content, duration, or pedagogical approaches. Given these constraints, our findings should be interpreted cautiously and validated through more comprehensive, multicenter longitudinal studies.

## 5. Conclusions

This study reveals significant gaps in endometriosis-related knowledge among nurses in the Al-Jouf region, highlighting educational attainment, clinical experience, and direct patient care as significant determinants. By situating these findings within the KAP model, the research underscores the importance of both theoretical and experiential learning pathways in enhancing clinical proficiency for women’s health issues. The insights affirm the pivotal role of nurse education but also challenge assumptions regarding the efficacy of existing training programs. Addressing these knowledge gaps has vital policy implications, whereby strategic investments in nurse education could accelerate diagnostic processes, improve patient outcomes, and mitigate the long-term burden of endometriosis.

## Figures and Tables

**Figure 1 healthcare-13-01386-f001:**
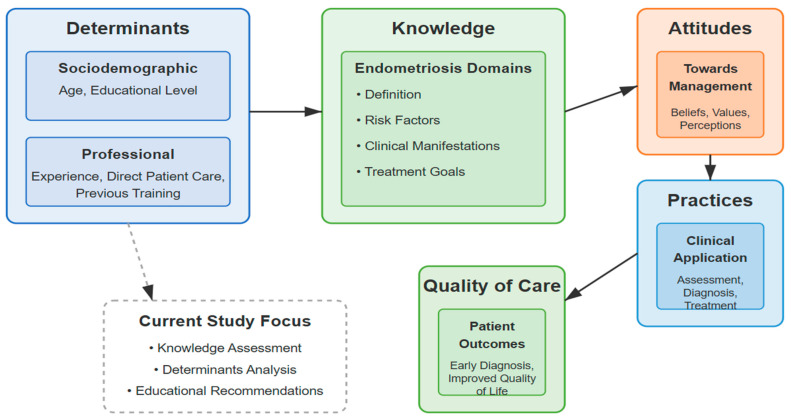
Conceptual Framework Based on the Knowledge-Attitude-Practice (KAP) Model. Note: This study examines only the knowledge component; attitudes and practices will be evaluated in subsequent phases.

**Table 1 healthcare-13-01386-t001:** Sociodemographic and professional characteristics of nurses (N = 215).

Characteristic	n	% (95% CI) *
Age (years) ^a^		
≤30	101	47.0 (40.3–53.7)
31–40	75	34.9 (28.8–41.6)
40–50	31	14.4 (10.1–19.7)
>50	8	3.7 (1.6–7.1)
Educational Level		
Diploma	69	32.1 (26.1–38.6)
Bachelor’s or higher +	146	67.9 (61.4–73.9)
Years of Experience		
<5	47	21.9 (16.7–27.9)
5–10	97	45.1 (38.4–51.8)
>10	71	33.0 (27.0–39.5)
Previous Endometriosis Training		
Yes	56	26.0 (20.5–32.1)
No	159	74.0 (67.9–79.5)
Direct Patient Care Experience with Endometriosis		
Yes	23	10.7 (7.0–15.5)
No	192	89.3 (84.5–93.0)

Note * 95% CI calculated using Wilson method for binomial proportions, + bachelor’s (n = 141) and master’s (n = 5) degrees combined. ^a^ Mean age = 32.1 ± 8.3 years, range 23–57 years.

**Table 2 healthcare-13-01386-t002:** Association between knowledge level and sociodemographic/professional characteristics (N = 215).

Characteristic	Total Sample n (%)	Satisfactory n (% of 84)	Unsatisfactory n (% of 131)	χ^2^	*p*-Value	Cramer’s V
Age (years)				11.175	0.011	0.228
≤30	101 (47.0)	28 (33.3)	73 (55.7)			
31–40	75 (34.9)	35 (41.7)	40 (30.5)			
41–50	31 (14.4)	16 (19.0)	15 (11.5)			
>50	8 (3.7)	5 (6.0)	3 (2.3)			
Educational Level				39.942	<0.001	0.431
Diploma	69 (32.1)	8 (9.5)	61 (46.6)			
Bachelor’s/Master’s	146 (67.9)	76 (90.5)	70 (53.4)			
Years of Experience				11.579	0.003	0.232
<5 years	47 (21.9)	13 (15.5)	34 (26.0)			
5–10 years	97 (45.1)	32 (38.1)	65 (49.6)			
>10 years	71 (33.0)	39 (46.4)	32 (24.4)			
Direct Patient Care with Endometriosis				16.618	<0.001	0.278
Yes	23 (10.7)	18 (21.4)	5 (3.8)			
No	192 (89.3)	66 (78.6)	126 (96.2)			
Previous Endometriosis Training				6.689	0.010	0.176
Yes	56 (26.0)	30 (35.7)	26 (19.8)			
No	159 (74.0)	54 (64.3)	105 (80.2)			

Notes: χ^2^ = Pearson’s chi-square; Cramer’s V indicates effect size (<0.2 = weak, 0.2–0.3 = moderate, >0.3 = strong). Knowledge groups: unsatisfactory (<60%) vs. satisfactory (≥60%) based on Bloom’s taxonomy-derived cut-offs; percentages in “Total Sample” column are of N = 215, and percentages in knowledge columns are within each knowledge group (n = 131 and n = 84, respectively).

**Table 3 healthcare-13-01386-t003:** Mean endometriosis knowledge domain scores by educational level (diploma vs. bachelor’s/master’s) and clinical experience (< 5 vs. ≥ 10 years).

Knowledge Domain	Educational Level	Years of Experience
Definition of Endometriosis (0–2) ^~^	*p* < 0.001 *	*p* = 0.037 *
- Low (reference)	0.82 ± 0.48 ‡	0.79 ± 0.45 ‡
- High	1.24 ± 0.52 **	1.31 ± 0.49 **
Risk Factors (0–4) ^~^	*p* < 0.001 *	*p* = 0.042 *
- Low (reference)	1.85 ± 0.92 ‡	1.78 ± 0.89 ‡
- High	2.46 ± 0.97 **	2.52 ± 0.95 **
Clinical Manifestations (0–10) ^~^	*p* < 0.001 *	*p* = 0.028 *
- Low (reference)	5.92 ± 1.68 ‡	5.85 ± 1.72 ‡
- High	7.34 ± 1.75 **	7.48 ± 1.70 **
Treatment Goals (0–4) ^~^	*p* < 0.001 *	*p* = 0.035 *
- Low (reference)	1.95 ± 0.82 ‡	1.88 ± 0.85 ‡
- High	2.48 ± 0.89 **	2.56 ± 0.87 **
Total Knowledge Score (0–20) ^~^	*p* < 0.001 *	*p* = 0.031 *
- Low (reference)	10.54 ± 2.93 ‡	10.30 ± 2.89 ‡
- High	13.87 ± 3.15 **	13.87 ± 3.08

Notes: Values presented as mean ± SD/^~^ possible score range for each domain/‡ reference groups: educational level (diploma), years of experience (<5 years)/** high groups: educational level (bachelor’s/master’s), years of experience (>10 years). * Statistically significant at *p* < 0.05 using one-way ANOVA. Educational levels combined due to small sample size in master’s category (n = 5).

**Table 4 healthcare-13-01386-t004:** Predictors of nurses’ knowledge about endometriosis: multiple linear regression analysis (N = 215).

Predictor Variables	B	SE	β	VIF	t	*p*-Value	95% CI	sr^2^
Constant	42.156	2.431	-	-	17.341	<0.001	37.365, 46.947	-
Age (years)	3.106	0.798	0.232	1.45	3.894	<0.001	1.534, 4.679	0.089
Educational Level (ref: Diploma)								
Bachelor’s or Higher	9.769	1.353	0.415	1.32	7.223	<0.001	7.103, 12.435	0.176
Experience (Years)	1.871	0.910	0.149	1.38	2.057	0.041	0.078, 3.665	0.042
Direct Patient Care ^1^	5.008	2.154	0.168	1.21	2.325	0.021	0.762, 9.254	0.051
Previous Training ^2^	1.327	1.490	0.056	1.18	0.890	0.374	−4.265, 1.611	0.008

Model Statistics: R^2^ = 0.376; adjusted R^2^ = 0.361; F(5, 209) = 25.209, *p* < 0.001; Durbin–Watson = 1.892; Condition Index = 18.234; residuals normality (Shapiro–Wilk): W = 0.984, p = 0.173; homoscedasticity (Breusch–Pagan): χ^2^ = 3.241, p = 0.518; maximum VIF = 1.45. Notes: ^1^ Direct patient care and ^2^ previous training reference category = no; B = unstandardized coefficient; SE = standard error; β = standardized coefficient; VIF = Variance Inflation Factor; sr^2^ = squared semi-partial correlation; CI = confidence interval. Model assumptions met for normality (*p* > 0.05), multicollinearity (all VIF < 2), and homoscedasticity (*p* > 0.05).

**Table 5 healthcare-13-01386-t005:** Factors associated with nurses’ knowledge of endometriosis: binary logistic regression analysis (N = 215).

Predictor Variables	B	SE	Wald χ^2^	*p*-Value	aOR	95% CI	VIF
Age (Years)	−0.319	0.195	2.667	0.103	0.727	0.495, 1.066	1.42
Educational Level ^1^	−1.897	0.419	20.524	<0.001	0.150	0.066, 0.342	1.35
Years of Experience	−0.408	0.226	3.258	0.071	0.665	0.427, 1.035	1.38
Direct Patient Care ^2^	−1.419	0.566	6.284	0.012	0.242	0.080, 0.734	1.24
Previous Training ^3^	−0.581	0.365	2.537	0.111	0.559	0.273, 1.143	1.19
Constant	2.865	0.451	40.326	<0.001	17.549	-	-

Model Fit: −2 Log likelihood = 245.326; Cox and Snell R^2^ = 0.289; Nagelkerke R^2^ = 0.392; Hosmer–Lemeshow test: χ^2^(8) = 6.784, p = 0.560; classification accuracy = 76.3%. Notes: B = regression coefficient; SE = standard error; aOR = adjusted odds ratio; CI = confidence interval; VIF = Variance Inflation Factor. 1 Educational level (0 = diploma, ^1^ = -bachelor’s or higher); ^2^ direct patient care (0 = no, 1 = yes); ^3^ previous training (0 = no, 1 = yes). Dependent variable: 0 = unsatisfactory knowledge, 1 = satisfactory knowledge. Model validated through bootstrapping (1000 samples).

**Table 6 healthcare-13-01386-t006:** Hierarchical multiple regression analysis of endometriosis knowledge determinants (N = 215).

Variables	Model 1	Model 2	Model 3
Step 1: Demographics			
- Age	0.232 (0.124, 0.340) **	0.185 (0.077, 0.293) *	0.168 (0.060, 0.276) *
Step 2: Educational Background			
- Educational Level	-	0.415 (0.307, 0.523) ***	0.382 (0.274, 0.490) ***
- Years of Experience	-	0.149 (0.041, 0.257) *	0.135 (0.027, 0.243) *
Step 3: Professional Exposure			
- Direct Patient Care	-	-	0.168 (0.060, 0.276) *
- Previous Training	-	-	0.056 (−0.052, 0.164)
Model Statistics			
- R^2^	0.124	0.298	0.376
- ΔR^2^	0.124	0.174	0.078
- F for ΔR^2^	15.012 ***	25.845 ***	12.654 ***
- Adjusted R^2^	0.116	0.285	0.361

Notes: Values are standardized coefficients β (95% CI)/* *p* < 0.05, ** *p* < 0.01, *** *p* < 0.001/CI = confidence interval/ΔR^2^ = change in R^2^/knowledge score range: 0–20.

## Data Availability

The datasets generated and analyzed during the current study are not publicly available to protect participant confidentiality. De-identified data may be requested from the corresponding author, subject to ethical approval and a signed data-use agreement.

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
