# Peer review of "Assessment of Endometriosis Knowledge and Its Determinants Among Nurses in Al-Jouf Region, Saudi Arabia"

_healthcare, 2025, doi:10.3390/healthcare13121386_

Round 1

Reviewer 1 Report

Comments and Suggestions for Authors

I read with interest the study presented by Elsharkawy and colleagues on the assessment of endometriosis knowledge and its determinants among nurses and its implications on the quality of care.

The presented study falls within the scope of the journal, and is of interest to the journal's readership. 

The manuscript presents a rather important topic that is often overlooked, yet of clinical significance. 

While the premise of the study warrants its publication, the study in its current form is riddled with significant issues that require authors' attention:

  • The title is grossly misleading and does not reflect the content of the study, specifically, the "Quality-Care Implications". The study only examined the determinants of nurses' knowledge without any assessment of the impact of such knowledge on the quality of care delivered by these nurses.
  • Replace the word "leading" in the abstract with "principal or main".
  • Authors are advised to use a neutral tone and avoid judgemental terminology such as "somber".
  • The introduction is unnecessarily long. Authors could significantly shorten the introduction and make it straight to the point. 
  • Authors should avoid single-word subheadings, e.g., Design or Settings. 
  • Authors should reference original sources of their used methodology, e.g, Bloom's cut-off.
  • Authors should clearly state their inclusion and exclusion criteria of their study participants.
  • When stating the IRB ethical approvals, authors should state the issuing authority, approval date and number.
  • As they have done with the Methods section, authors are encouraged to divide their results into subsection and present them in a logical order. 
  •  Authors should start their results by describing their study cohorts, e.g., gender, age, educational level, location of practice, years of experience, nationality ...etc. These descriptive statistics should be presented before starting comparing between the different subgroups.
  • It is unclear why authors have collected age in as a continuous variable, presented it as mean with standard deviations, and then grouped participants into age categories, and on what basis was this categorisation done.
  • Figure 2 is redundant and can be easily replaced with a single sentence.
  • To ensure consistency in reporting the results, authors are advised to adhere to a single categorisation of nurses' level of education, either (low, intermediate and high) or (unsatisfactory and satisfactory). Otherwise, the presentation/interpretation of their results is biased and do not necessarily reflect their findings. 
  • The presentation of the results is below publication standards. Figures and tables should supplement and support the reporting of the results, not the other way around. In other words, authors should describe their findings, and use the tables as a reference for the entire data, but not start their results by "Table 2 presents ..).  
  • Similar to the introduction, the discussion section is incredibly long and overstate the study findings. Authors should significantly reduce the two subsection included "Practical Implications" and "Future Research Direction"
  • Although the authors have attempted to discuss their study limitations in a great depth, it is unacceptable to include 27 lines worth of limitations without a single reference to support their argument.
  • Moreover, a study of this magnitude with such limited findings does not warrant 17-line conclusion. Authors should limit their conclusion to reflect their study findings without exaggeration.  

Author Response

Comment (1) :
The title is grossly misleading and does not reflect the content of the study, specifically, the "Quality-Care Implications". The study only examined the determinants of nurses' knowledge without any assessment of the impact of such knowledge on the quality of care delivered by these nurses.

response (1) :
We appreciate the reviewer’s point of view, and the title of the paper was changed to better represent the idea. The new title is: “Assessing Endometriosis Knowledge Among Nurses in the Al-Jouf Region of Saudi Arabia.”

"Assessment of Endometriosis Knowledge and its Determinants Among Nurses in Al-Jouf Region, Saudi Arabia.

Comment (2) :
Replace the word "leading" in the abstract with "principal or main".

response(2) :
As suggested, we have replaced the word "leading" with "principal" in the abstract to improve precision and clarity of language.

Comment (3) :
Authors are advised to use a neutral tone and avoid judgmental terminology such as "somber".

response(3) :
We appreciate the reviewer's guidance regarding the use of a neutral tone. We have replaced the term "somber" with more neutral descriptive language, "concerning."

Comment (4) :
The introduction is unnecessarily long. Authors could significantly shorten the introduction and make it straight to the point.

response(4) :
We acknowledge that the introduction was unnecessarily lengthy. We have significantly condensed this section while maintaining essential background information on endometriosis prevalence, impact, and the critical role of nurses in its management.

Comment (5) :
Authors should avoid single-word subheadings, e.g., Design or Settings

response(5) :
We have revised all single-word subheadings throughout the manuscript. Specifically: – "Design" has been changed to "Study Design" – "Settings" has been changed to "Study Settings".

Comment (6) :
Authors should reference original sources of their used methodology, e.g, Bloom's cut-off.

response(6) :
We have added appropriate references for all methodological approaches used in our study. Specifically, we have included the original source for Bloom's cut-off criteria (Bloom et al., 1956) and added it to the list of references.

Comment (7) :
Authors should clearly state their inclusion and exclusion criteria of their study participants.

response(7) :
We have clearly articulated the inclusion and exclusion criteria for study participants in the Methods section.

Inclusion criteria encompassed registered nurses actively involved in direct patient care in selected healthcare facilities, including exposure to patients with endometriosis, with a minimum of one year of clinical experience, and willing to provide written informed consent. Exclusion criteria included nurses who held solely administrative roles, had participated in similar studies previously, were unable to complete the questionnaire due to physical or psychological impairments, or maintained a personal relationship with women diagnosed with endometriosis, factors that could potentially bias their responses.

Page # 5, lines 190-197.

Comment (8) :
When stating the IRB ethical approvals, authors should state the issuing authority, approval date, and number.

response(8) :
We have revised the ethical approval statement to include the issuing authority, approval date, and number as follows: "This study received approval from the Local Committee of Bioethics at Jouf University (IRB number: 7-04-45, dated December 03, 2023) and the Ministry of Health, Research Ethics Committee, Qurayyat Health Affairs (IRB Approval No. 2023-121, dated December 10. 2023)."

Page # 8, lines 312-315.

comment (9) :
As they have done with the Methods section, authors are encouraged to divide their results into subsections and present them in a logical order.

response(9) :
Following the reviewer's suggestion, we have reorganized the Results section into logical subsections:
3.1. Demographic Characteristics of Study Participants.
3.2. Knowledge Assessment Scores.
3.3. Factors Associated with Knowledge Levels.
3.4. Multivariate Analysis of Knowledge Determinants.

comment (10) :
Authors should start their results by describing their study cohorts, e.g., gender, age, educational level, location of practice, years of experience, nationality ... etc. These descriptive statistics should be presented before starting comparing between the different subgroups.

response(10) :
We have moved the descriptive statistics of our study cohort to the beginning of the Results section, providing a comprehensive overview of participants' age, educational level, and years of experience, before presenting comparative analyses.

comment (11) :
It is unclear why authors have collected age in as a continuous variable, presented it as mean with standard deviations, and then grouped participants into age categories, and on what basis was this categorisation done.

response(11) :
Thank you for your thoughtful comment regarding our approach to age data collection and presentation. We would like to clarify our methodological decisions regarding the age variable in our study.

We collected age as a continuous variable to maximize statistical precision in our regression analyses, where age was a significant predictor (β = 0.232, p < 0.001) of endometriosis knowledge. Presenting age as mean with standard deviation follows standard epidemiological practice for continuous variables.

The subsequent categorization into age groups served two purposes: to facilitate the interpretation of knowledge levels across different career stages and to enable subgroup analyses that might reveal non-linear patterns not captured in regression models. Our categorization was guided by developmental career stages in nursing practice while ensuring sufficient sample sizes within each group for valid statistical comparisons.

This dual approach to analyzing age aligns with our Knowledge-Attitude-Practice framework, which examines how experiential learning influences clinical knowledge acquisition.

comment (12) :
Figure 2 is redundant and can be easily replaced with a single sentence.

response(12) :
We agree with the reviewer's assessment that Figure 2 was redundant. We have removed this figure and replaced it with a concise textual description of the findings it previously illustrated.

comment (13) :
To ensure consistency in reporting the results, authors are advised to adhere to a single categorisation of nurses' level of education, either (low, intermediate and high) or (unsatisfactory and satisfactory). Otherwise, the presentation/interpretation of their results is biased and do not necessarily reflect their findings.

response(13) :
The classification of education level in our study aligns with the professional qualifications of the nurses, categorized as Diploma or Bachelor’s/Master’s. In contrast, nurses’ knowledge levels were assessed as either Unsatisfactory (< 60 %) or Satisfactory (≥ 60 %) according to Bloom’s taxonomy-based thresholds.

comment (14) :
The presentation of the results is below publication standards. Figures and tables should supplement and support the reporting of the results, not the other way around. In other words, authors should describe their findings and use the tables as a reference for the entire data, but not start their results by "Table 2 presents ...".

response(14) :
We have thoroughly revised our presentation of results to meet publication standards. Rather than beginning sections with references to tables or figures, we now describe our findings in the text and use tables and figures as supplementary references.

You can find that in the updated version of the results section of the manuscript, pages 9-14, lines 398-523.

comment (15) :
Similar to the introduction, the discussion section is incredibly long and overstates the study findings. Authors should significantly reduce the two subsection included "Practical Implications" and "Future Research Direction".

response(15) :
We have significantly condensed "Practical Implications" and "Future Research Direction" subsections, while maintaining the essential insights and recommendations. The revised discussion is more focused on interpreting our findings in the context of existing literature and their implications for nursing practice and education.

Page # 17, lines 657-674.

comment (16) :
Although the authors have attempted to discuss their study limitations in a great depth, it is unacceptable to include 27 lines worth of limitations without a single reference to support their argument.

response(16) :
We have revised the limitations section to be more concise and have added appropriate references to support our discussion of methodological limitations. The section now includes citations to relevant literature on cross-sectional study limitations, convenience sampling challenges, and knowledge assessment methodologies.

The present study has several methodological limitations. The cross-sectional design precludes causal inferences between educational factors and knowledge levels [70]. The convenience sampling from three health centers in Al-Jouf limits generalizability, though it provides valuable preliminary insights from this understudied region [71]. While the knowledge assessment tool was originally designed for undergraduate students, we validated it for practicing nurses (CVI = 0.93). The application of the KAP model focused solely on knowledge, with attitude and practice dimensions planned for subsequent research. The unexpected finding that previous training did not significantly impact knowledge levels warrants further investigation, as we did not collect detailed information about training content, duration, or pedagogical approaches. Given these constraints, our findings should be interpreted cautiously and validated through more comprehensive, multicenter longitudinal studies.

Page 17-18, lines 676-687.

comment (17) :
Moreover, a study of this magnitude with such limited findings does not warrant a 17-line conclusion. Authors should limit their conclusions to reflect their study findings without exaggeration.

response(17) :
We have shortened the conclusion to more accurately reflect our study findings without exaggeration. The revised conclusion is focused on the key findings regarding nurses' knowledge of endometriosis and their implications for nursing education and practice in the Al-Jouf region.

This study reveals significant gaps in endometriosis-related knowledge among nurses in the Al-Jouf region, highlighting educational attainment, clinical experience, and direct patient care as significant determinants. By situating these findings within the KAP model, the research underscores the importance of both theoretical and experiential learning pathways in enhancing clinical proficiency for women’s health issues. The insights affirm the pivotal role of nurse education but also challenge assumptions regarding the efficacy of existing training programs. Addressing these knowledge gaps has vital policy implications, whereby strategic investments in nurse education could accelerate diagnostic processes, improve patient outcomes, and mitigate the long-term burden of endometriosis.

Page # 18, line 689-697.

Reviewer 2 Report

Comments and Suggestions for Authors

When the article is examined carefully, some important limitations and deficiencies are noted despite its methodological strengths. First of all, although the cross-sectional design of the study is convenient in explaining the relationships between knowledge levels and sociodemographic variables, it does not allow for establishing a causal relationship. This situation should be considered carefully, for example, when questioning whether education really increases knowledge levels. In addition, the sample selection was made by convenience sampling and only three health centers were focused on; this situation reduces the representativeness of the sample and limits the generalizability of the findings. Although the scale validation process is explained in detail, the fact that the knowledge test used was originally designed for undergraduate students may raise some questions about its validity in assessing the knowledge level of working nurses. In addition, although the KAP (Knowledge-Attitude-Practice) model was adopted as the theoretical framework in the study, only the “knowledge” component was focused on and the “attitude” and “practice” dimensions were excluded from the evaluation; this situation prevented the holistic use of the model and reduced conceptual depth. Finally, although the effect of education level and experience on knowledge level has been shown statistically, the ineffectiveness of previous trainings could not be explained sufficiently due to the lack of sufficient information about the content, duration and implementation method of these trainings. These deficiencies indicate that the results of the study should be interpreted with caution and that more comprehensive, multicenter, longitudinal studies are needed in the future.

Author Response

Comment (1) :
When the article is examined carefully, some important limitations and deficiencies are noted despite its methodological strengths. First of all, although the cross-sectional design of the study is convenient in explaining the relationships between knowledge levels and sociodemographic variables, it does not allow for establishing a causal relationship. This situation should be considered carefully, for example, when questioning whether education really increases knowledge levels. In addition, the sample selection was made by convenience sampling and only three health centers were focused on; this situation reduces the representativeness of the sample and limits the generalizability of the findings. Although the scale validation process is explained in detail, the fact that the knowledge test used was originally designed for undergraduate students may raise some questions about its validity in assessing the knowledge level of working nurses. In addition, although the KAP (Knowledge-Attitude-Practice) model was adopted as the theoretical framework in the study, only the “knowledge” component was focused on and the “attitude” and “practice” dimensions were excluded from the evaluation; this situation prevented the holistic use of the model and reduced conceptual depth. Finally, although the effect of education level and experience on knowledge level has been shown statistically, the ineffectiveness of previous trainings could not be explained sufficiently due to the lack of sufficient information about the content, duration and implementation method of these trainings. These deficiencies indicate that the results of the study should be interpreted with caution and that more comprehensive, multicenter, longitudinal studies are needed in the future.

response(1) :
We acknowledge the reviewer’s valid observation regarding the cross-sectional design of our study. We agree that while this design effectively establishes relationships between knowledge levels and sociodemographic variables, it does not permit causal inferences. We have added this important limitation, specifically noting the following:

The present study has several methodological limitations. The cross-sectional design precludes causal inferences between educational factors and knowledge levels [70]. The convenience sampling from three health centers in Al-Jouf limits generalizability, though it provides valuable preliminary insights from this understudied region [71]. While the knowledge assessment tool was originally designed for undergraduate students, we validated it for practicing nurses (CVI = 0.93). The application of the KAP model focused solely on knowledge, with attitude and practice dimensions planned for subsequent research. The unexpected finding that previous training did not significantly impact knowledge levels warrants further investigation, as we did not collect detailed information about training content, duration, or pedagogical approaches. Given these constraints, our findings should be interpreted cautiously and validated through more comprehensive, multicenter longitudinal studies.

Page 17 – 18, lines 676-687.

Reviewer 3 Report

Comments and Suggestions for Authors

The study makes a significant contribution to understanding the level of knowledge about endometriosis among nurses and highlights important determinants for continuing education in this field. It may be useful to include an additional discussion regarding the limitations of the knowledge assessment tool — even though it was validated, specifying the original source and the type of questions included would be of interest.

A brief comparison with similar studies conducted in other regions or countries could also be added in the discussion section to highlight the international relevance of the findings.

Comments on the Quality of English Language

A final linguistic revision could improve the fluency of certain phrases, particularly in the discussion section.

Author Response

Comment (1) :

The study makes a significant contribution to understanding the level of knowledge about endometriosis among nurses and highlights important determinants for continuing education in this field. It may be useful to include an additional discussion regarding the limitations of the knowledge assessment tool, even though it was validated, specifying the original source and the type of questions included would be of interest.

response(1) :

We sincerely thank the reviewer for this valuable suggestion. In response, we have added a dedicated paragraph to the discussion section elaborating on the limitations of the knowledge assessment tool.

Despite rigorous psychometric validation, the endometriosis knowledge tool used in this study has inherent limitations. Originally developed by Schlorke (2021) [25] for U.S. undergraduate nursing students, it may not fully reflect the clinical depth or regional nuances of endometriosis care in Saudi Arabia. Although adapted and validated for the Arabic-speaking population, including a four-domain structure (definition, risk factors, clinical manifestations, and treatment goals), the tool focuses solely on factual knowledge. It may underrepresent nurses’ critical thinking, diagnostic awareness, or patient communication abilities.

Page # 17, lines 636-643

Comment (2) :

A brief comparison with similar studies conducted in other regions or countries could also be added in the discussion section to highlight the international relevance of the findings.

response(2) :

We appreciate this insightful recommendation and have expanded the discussion to include comparisons with findings from other countries.

The findings regarding nurses' knowledge of endometriosis align with international trends and reveal regional variations. European studies reveal higher knowledge levels. Medina-Perucha et al. [58], found that 47% of Spanish nurses had satisfactory knowledge, in contrast to our 39%, which may be linked to specialised training programs. Bach et al [36], demonstrated that Danish nurses specialising in gynecology had a higher recognition of endometriosis symptoms than the general nursing sample in our study. In contrast, Middle Eastern studies demonstrate patterns consistent with our findings. Hassan [59], reported that 65% of Egyptian nurses demonstrated low knowledge scores, while Al-Jefout et al. [60], found that only 34% of Jordanian nurses possessed a satisfactory understanding of endometriosis. Regional similarities suggest that cultural factors and educational methodologies in the Middle East and North Africa (MENA) region contribute to the presence of comparable knowledge gaps. This international comparison indicates that while knowledge deficits concerning endometriosis among nurses are a widespread issue, their severity varies across different healthcare systems. This study provides insights from a previously unexamined region of Saudi Arabia, emphasising the need for educational interventions tailored to the specific context, guided by successful international models.

Page # 16, lines 585-600

Comment (3) :
Comments on the Quality of English Language
A final linguistic revision could improve the fluency of certain phrases, particularly in the discussion section.

response(3) :

Thank you for pointing this out. We have conducted a thorough review of language and grammar throughout the discussion section, with a particular focus on sentence fluency, clarity, and conciseness. Ambiguous or overly complex phrasing has been simplified, and transitional sentences have been added where appropriate to enhance the logical flow. We also used Grammarly Premium and cross-reviewed the edits with a bilingual academic colleague to ensure both technical accuracy and natural English expression.

Round 2

Reviewer 1 Report

Comments and Suggestions for Authors

Congratulations to the authors for their efforts in addressing most of the comments raised in the first round of revision.

The manuscript has been significantly improved as a result of such modifications. However, a few minor issues remain:

  • Some subsections in the methods are unnumbered.
  • It seems that the authors were confused about a previous comment. Tables/Figures should come AFTER their description and reference in the text, not the other way around.

Author Response

Comment 1 

Congratulations to the authors for their efforts in addressing most of the comments raised in the first round of revision.

The manuscript has been significantly improved as a result of such modifications. However, a few minor issues remain:

Some subsections in the methods are unnumbered.

It seems that the authors were confused about a previous comment. Tables/Figures should come AFTER their description and reference in the text, not the other way around

Response 1

We sincerely thank you for the positive feedback and for acknowledging the substantial improvements made in the revised manuscript. We appreciate your constructive observations and have addressed the remaining issues as follows:

1-Unnumbered Subsections in the Methods Section

Thank you for this observation. We have carefully reviewed the “Materials and Methods” section and ensured that all subsections are now consistently numbered to align with the journal’s formatting guidelines. The numbering sequence has been applied throughout to facilitate clarity and navigability.

Required Change in the File:

Go to the "2. Materials and Methods" section. You need to add subsection numbers to the currently unnumbered items. Suggested numbering scheme:

2.1 Study Design

2.2 Study Setting

2.3 Study Participants and Sample Size Determination

2.3.1. Variable Selection Strategy 

2.3.2. Inclusion and Exclusion Criteria 

2.3.3. Sample Size Calculation 

2.4 Data Collection Tools

2.4.1. Instrument Overview 

2.4.2. Validation and Reliability Procedures 

2.5 Ethical Approval

2.6 Procedure

2.7 Statistical Analysis

2- Placement of Tables and Figures

We have now revised the manuscript so that all Tables and Figures appear after their first mention in the text, as per journal formatting requirements. The necessary changes have also been made in the attached manuscript file.

We trust that these revisions fully address the remaining concerns. Thank you again for your thoughtful review and guidance throughout the revision process.

Reviewer 2 Report

Comments and Suggestions for Authors

The article has been sufficiently revised and is acceptable.

Author Response

comment 1 
The article has been sufficiently revised and is acceptable.

response 1 
thank you